# Autoxidation Enhances Anti-Amyloid Potential of Flavone Derivatives

**DOI:** 10.3390/antiox10091428

**Published:** 2021-09-07

**Authors:** Andrius Sakalauskas, Mantas Ziaunys, Ruta Snieckute, Vytautas Smirnovas

**Affiliations:** Life Sciences Center, Institute of Biotechnology, Vilnius University, LT-10257 Vilnius, Lithuania; andrius.sakalauskas@gmc.vu.lt (A.S.); mantas.ziaunys@gmail.com (M.Z.); r.snieckute@gmail.com (R.S.)

**Keywords:** aggregation, amyloid-beta, insulin, flavones, inhibition, autoxidation

## Abstract

The increasing prevalence of amyloid-related disorders, such as Alzheimer’s or Parkinson’s disease, raises the need for effective anti-amyloid drugs. It has been shown on numerous occasions that flavones, a group of naturally occurring anti-oxidants, can impact the aggregation process of several amyloidogenic proteins and peptides, including amyloid-beta. Due to flavone autoxidation at neutral pH, it is uncertain if the effective inhibitor is the initial molecule or a product of this reaction, as many anti-amyloid assays attempt to mimic physiological conditions. In this work, we examine the aggregation-inhibiting properties of flavones before and after they are oxidized. The oxidation of flavones was monitored by measuring the UV-vis absorbance spectrum change over time. The protein aggregation kinetics were followed by measuring the amyloidophilic dye thioflavin-T (ThT) fluorescence intensity change. Atomic force microscopy was employed to image the aggregates formed with the most prominent inhibitors. We demonstrate that flavones, which undergo autoxidation, have a far greater potency at inhibiting the aggregation of both the disease-related amyloid-beta, as well as a model amyloidogenic protein—insulin. Oxidized 6,2′,3′-trihydroxyflavone was the most potent inhibitor affecting both insulin (7-fold inhibition) and amyloid-beta (2-fold inhibition). We also show that this tendency to autoxidize is related to the positions of the flavone hydroxyl groups.

## 1. Introduction

Protein aggregation into highly structured aggregates is associated with various amyloidoses, such as Alzheimer’s disease (AD) and Parkinson’s disease (PD) [1]. AD alone is recognized to be the most common cause of dementia (60–80%) [2] that affects more than 50 million people worldwide and, according to the World Alzheimer’s Report, is set to increase up to 152 million by 2050. The cause of this forecast is that the onset of AD mostly occurs after 60 years of age, and the increasing life expectancy leads to more people suffering from dementia. The pathological hallmark of this disease is the increased concentration of the 42 amino acid peptide—amyloid-beta (Aβ_42_) that prompts the formation of its oligomeric and fibrillar species [3].

The increasing focus on anti-amyloid-β compounds has led to many different in vitro studies showing positive effects against protein aggregation [4]. Despite this fact, many suggested disease-modifying compounds, ranging from small organic molecules to large monoclonal antibodies, have not led to an effective cure, leaving 99.5% of clinical trials unsuccessful [5,6]. Several potential problems with the very low clinical trial success rate are linked to targeting the wrong pathological substrates, concerns with drug development, and problems with methodologies [7,8]. Subsequently, it is of utmost importance to take into consideration the gap between the initial drug screening and human physiology [4,9].

The aggregation process of the Aβ_42_ peptide is exceptionally complicated; however, the mechanism is rather well described [10,11]. The process of several steps involves primary nucleation, elongation, fibril surface-catalyzed nucleation (often referred to as secondary nucleation), and fragmentation [12]. While primary nucleation causes the formation of nuclei that eventually grow into fibril aggregates, secondary nucleation is shown to be the main source of more cytotoxic oligomeric species that cause direct neurotoxicity [13,14,15]. For that reason, it is beneficial to find an anti-amyloidogenic compound that prevents primary and secondary nucleation as well as elongation processes [16].

Flavones are abundant in nature and found in a variety of herbs, fruits, vegetables, and spices [17]. This group of natural anti-oxidants has been reported to possess anti-amyloid characteristics, exhibit neuroprotective, anti-inflammatory, and anti-microbial properties [16,18]. In addition, flavone derivatives have shown positive effects when treating diabetes, cancer, malaria, asthma, and cardiovascular system diseases [19]. Studies have also shown that a variety of flavonoids function as acetylcholinesterase inhibitors (AChEI) [20,21]. AChEI is currently one of the most prominent options for symptomatic treatment of AD, mostly by increasing neurotransmitter acetylcholine concentrations in synaptic gaps of the nervous system [22,23]. If the same compound would also inhibit amyloid formation, it could be an ultimate anti-amyloid drug. Moreover, the small molecular weight and widely abundant flavonoids could be a better option for drug development. Compared to the large monoclonal antibody-based drugs, such molecules do pass Lipinski’s rule of 5, have high availability and stability, and could potentially be used for less expensive prevention against the onset of neurodegenerative diseases [24].

Studies with flavones demonstrated properties against Aβ_42_ aggregation in vitro [25,26]. In many cases, the anti-aggregation potential is evaluated via measurement of amyloidophilic dye thioflavin-T (ThT) fluorescence intensity [27,28], assuming that relatively lower fluorescence intensity correlates with fewer fibrils formed. While this hypothesis is quite prominent, various counterfactors exist. Typically, Aβ_42_ aggregation is examined at neutral pH without evaluating the characteristics of the potential inhibitor in question. Numerous flavones have light absorbance properties in the same range as typically used fluorescent amyloid-dyes [29]. In addition, flavones could potentially bind to either the dye molecule itself or the formed aggregates, preventing its interaction with the fibril [30].

Many polyphenolic compounds, including flavones, are reported to undergo autoxidation at neutral or higher pH [31,32]. One particular study shows the oxidation mechanism of quercetin, suggesting that the process involves the breakdown of the flavone C ring, enabling different structure formations [32]. In another report, the Aβ_42_ inhibitory effect is based on the autoxidation of (+)-taxifolin [28]. This leads to an assumption that molecule autoxidation could be the main cause of the inhibitory effect in vitro. Furthermore, several reports demonstrate low mono- and polyhydroxylated flavone oral bioavailability due to direct metabolism [33]. In addition, human cytochrome P450 enzymes oxidize the 5-hydroxyflavone to specific di- or trihydroxyflavones [34]. These aforementioned aspects raise questions about whether the tested molecule or its oxidized species inhibit amyloid formation in vitro.

In this work, we examined the oxidation potential of 64 mono- and polyhydroxylated flavones and tested their inhibitory effect on the aggregation of amyloid-beta and a commonly used model amyloid protein—insulin. We show that the positions of flavone hydroxy groups have a remarkably high impact on autoxidation which enables the inhibitory effect on both proteins under the tested conditions.

## 2. Materials and Methods

### 2.1. Flavone Solution Preparation

Each non-oxidized flavone stock solution was prepared by dissolving the flavones (Indofine Chemical Company, Inc., Hillsborough, NJ, USA) in dimethylsulfoxide (DMSO, Carl Roth, Karlsruhe, Germany) to a final concentration of 10 mM. The oxidation solution of each flavone was prepared by diluting 10 mM flavone stock solution with 10 mM sodium phosphate buffer (pH 8.0) and DMSO to yield a final flavone concentration of 0.2 mM in 9 mM sodium phosphate buffer solution containing 10% DMSO. The 10% DMSO buffer solution was used to increase the solubility of flavones.

### 2.2. Absorbance Measurements

The autoxidation of flavones was monitored by measuring UV-Vis absorbance spectrum changes over time using a ClarioStar Plus plate reader (BMG Labtech, Ortenberg, Germany). Each flavone oxidation solution was stored as 100 µL samples in a UV-clear 96-well plate (Thermo Fisher Scientific, Inc., Waltham, MA, USA, cat. No. 11670352) and incubated at 37 °C, while the measuring absorbance spectra were in the range from 240 nm to 800 nm. Data were collected each hour for a total of 100 h. Spectra was baseline corrected at 800 nm. The resulting samples, which are later referred to as “incubated” or “oxidized” flavones, were then used in aggregation kinetic experiments.

Samples for the measurement of ThT and flavone interaction were prepared by mixing 0.5 mM incubated flavone, 10 mM ThT stock solution, and 20 mM phosphate buffer solution (pH 7.0), yielding either separate 50 µM flavone and 20 µM ThT or combined 50 µM flavone and 20 µM ThT solutions in 20 mM phosphate buffer (pH 7.0). Samples were scanned using a Shimadzu UV-1800 spectrophotometer (1 nm steps). Separate 50 µM flavone and 20 µM ThT spectra were added together for comparison with their mixture. Each sample was scanned three times and averaged; the baseline was corrected at 800 nm.

### 2.3. Fluorescence Measurements

Samples for the fluorescence measurements were prepared by mixing 0.5 mM incubated flavone, 10 mM ThT stock solution, 2 µM of Aβ_42_ aggregates, and 20 mM phosphate buffer solution (pH 7.0), yielding 1 µM of Aβ_42_ fibril samples with either 20 µM ThT or 50 µM flavone samples with both ThT and flavone. The fluorescence intensity was scanned using a Varian Cary Eclipse fluorescence spectrophotometer, with excitation and emission wavelengths being 440 nm and 480 nm, respectively (5 nm excitation and 2.5 nm emission slit widths). The intrinsic fluorescence emission intensity, occurring from non-fibril-bound ThT or flavones, was subtracted from their respective fibril-compound sample intensities. This was done by acquiring fluorescence emission intensity values of ThT or flavone samples in the absence of Aβ_42_ aggregates.

### 2.4. Purification of Recombinant Aβ_42_

The expression vector of Aβ_42_ was described previously [35]. The peptide was expressed in *E.coli* BL-21Star^TM^ (DE3) (Invitrogen, Carlsbad, CA, USA) and purified as described previously [36]. In brief, the transformed cells were incubated on LB agar plates containing ampicillin (100 µg/mL) overnight at 37 °C. The next day, the overnight cultures were prepared from single colonies and grown in LB medium with ampicillin (100 µg/mL). The 1 mL of the culture was transferred to 400 mL of auto-inductive ZYM-5052 medium [37] containing ampicillin (100 µg/mL) and grown for 15 h. The collected cell pellet was washed 3 times to remove all soluble proteins. The procedure involves pellet homogenization, sonication, and centrifugation. After removing soluble proteins, the cell pellet was resuspended in 50 mL of 20 mM Tris/HCL pH 8.0 buffer solution containing 8 M urea and 1 mM EDTA, homogenized, and centrifuged as in the previous steps. The collected supernatant was diluted with 150 mL of 20 mM Tris/HCL (pH 8.0) buffer containing 1 mM EDTA, mixed with 60 mL DEAE-sepharose and agitated at 80 rpm for 30 min at 4 °C. The chromatography procedure was performed using a Buchner funnel with Fisherbrand glass microfiber paper on a vacuum glass bottle. The resin with bound proteins was washed with 20 mM Tris/HCL pH 8.0 buffer containing 1 mM EDTA in increasing NaCl concentrations in a step-gradient (0, 20, 150, 500 mM). The target protein fractions were collected by washing the resin with a 50 mL buffer solution (containing 150 mM NaCl) four times. Collected fractions were mixed together, lyophilized, and stored at −20 °C.

The Aβ_42_ peptide powder was dissolved in a 20 mM sodium phosphate buffer solution (pH 8.0) containing 5 M guanidine thiocyanate (GuSCN, Carl Roth). The dissolved sample was loaded on a Tricorn 10/300 column (packed with Superdex 75) and eluted at 1 mL/min using a 20 mM sodium phosphate buffer solution (pH 8.0) containing 0.2 mM EDTA and 0.02% NaN_3_. Collected fractions were mixed together, lyophilized, and stored at −20 °C. Before aggregation experiments, the purification procedure was repeated, but this time the collected fraction (0.75 mL) was purified Aβ_42_ was stored on ice for 5 min. The concentration was determined by calculating the integrated chromatographic UV absorbance peak (ε_280_ = 1 490 M^−1^ cm^−1^). Afterward, it was diluted and immediately used for aggregation experiments.

### 2.5. Aggregation Kinetics of Aβ_42_ Peptide

The purified peptide fraction (1.5 mL, pH 8.0) was mixed with 3 mL of 20 mM sodium phosphate buffer solution (pH 6.33) to yield a 3-fold diluted peptide solution (pH 7.0). The peptide and each oxidized or incubated flavone solution was mixed together with 20 mM sodium phosphate buffer solution (pH 7.0), 10 mM ThT stock solution, and DMSO to a final reaction mixture, containing 1 µM Aβ_42_, 20 µM ThT, 50 µM of selected flavone compound and 1% DMSO. The kinetic aggregation measurements were performed in non-binding 96-well plates (Fisher, Waltham, MA, USA, cat. No. 10438082) (sample volume was 80 µL) at 37 °C by measuring ThT fluorescence using 440 nm excitation and 480 emission wavelengths in a ClarioStar Plus (BMG Labtech, Ortenberg, Germany).

### 2.6. Aggregation Kinetics of Insulin

Human recombinant insulin powder (Sigma-Aldrich, St. Louis, MO, USA, cat. No. 91077C) was dissolved in a 20% acetic acid solution (prepared from 100% acetic acid; Carl-Roth) containing 100 mM NaCl (Fisher) to a protein concentration of 400 µM. This insulin stock solution was mixed with non-oxidized/incubated or oxidized/incubated flavone solutions and 10 mM ThT stock solution to a final insulin concentration of 200 µM, 100 µM ThT, and 20 µM of each flavone. The aggregation kinetic measurements were performed similarly as in the case of Aβ_42_, but at 60 °C.

### 2.7. Kinetic Data Analysis

After reaching the plateau, kinetic aggregation curves were fit using Boltzmann’s sigmoidal equation:(1)$$y=\frac{\left(A_{1}-A_{2}\right)}{\left(1+e^{\frac{x-x_{0}}{dx}}\right)}+A_{2}$$
where, *A*_1_ is the starting fluorescence intensity, *A*_2_—final fluorescence intensity, *x*_0_—aggregation halftime. The relative halftime and relative ThT fluorescence intensity values were calculated based on the control sample in their specific microplate. These values were calculated by dividing each sample’s average value by the average control value. Data were processed using Origin software (OriginLab, Northampton, MA, USA).

### 2.8. Atomic Force Microscopy (AFM)

The samples for AFM images were collected after kinetic measurements and scanned similarly as previously described [31,38]. In short, 40 µL of 1% (*v/v*) APTES (Sigma-Aldrich, cat. No. 440140) in MilliQ water was deposited on freshly cleaved mica and incubated for 5 min. Then, mica was rinsed with 2 mL of MilliQ water and dried under gentle airflow. Each sample was deposited (40 µL) on the functionalized surface and incubated for another 5 min. Prepared samples were rinsed with 2 mL of MilliQ water and dried under gentle airflow. AFM imaging was performed using a Dimension Icon (Bruker, Billerica, MA, USA) atomic force microscope. Images were 1024 × 1024 pixel resolution and were analyzed using Gwyddion 2.5.5 software. Fibril heights were determined by tracing perpendicular to each fibril’s axis.

### 2.9. FTIR

Aβ_42_ fibrils were separated from the buffer solution by placing the mixture in the 0.5 mL 10 kDa concentrators (Fisher, cat. No. 88513) and spinning at 10,000 g for 10 min. Then 0.5 mL of D_2_O was added, and the process of buffer exchange to D_2_O was repeated 3 times. After the last spinning step, fibrils were resuspended in 0.1 mL of D_2_O. FTIR spectra were recorded using an Invenio S IR spectrophotometer equipped with an MCT detector. The sample was placed in the CaF_2_ transmission windows with 0.05 mm Teflon spacers, 256 interferograms of 2 cm^−1^ resolution were averaged per spectrum. All spectra were normalized in the 1705–1595 cm^−1^ region, and baseline corrected after subtracting the D_2_O and water vapor spectrums. The data were processed using GRAMS software (Thermo Fisher Scientific, Inc., Waltham, MA, USA).

## 3. Results

We first incubated flavones at 37 °C in order to evaluate potential structural transitions that occur due to autoxidation. The time-dependent changes in the UV-vis spectra of flavones were recorded over a period of 100 h, comparing the absorbance in the 240–800 nm region. At the start of the experiment, each flavone spectrum (Figure 1) exhibited two characteristic maxima that are associated with the π → π* transitions within rings A and C, referred to as benzoyl system, band II (~240–290 nm), and ring B that is conjugated with the carbonyl of ring C, referred to as cinnamoyl system, band I (~300–415 nm) [39] (Appendix A). A decrease in the magnitude of these bands was observed in all displayed spectra that led to no characteristic maxima (No. 11, 22, 31, 38, 44, 46, 48, 51–52, 57, 59, 64) or appearance of new maxima peaks in other cases. The absorbance spectra changes and reduced characteristics of the band I indicate structural changes, loss of conjugation in a chromophore, and development of different intra- and intermolecular interactions [40]. A few trihydroxyflavones (THF) (No. 38, 46), tetrahydroxyflavones (TeHF) (51–52), and most of penta- and hexahydroxyflavones (PHF and HHF) (No. 59, 61, 63, 64) had major spectrum changes within the first 5 h. Most of the other flavones, including dihydroxyflavones (DHF), THF, and TeHF (No.10, 11, 22, 31, 38, 42, 44, 46, 48, 53, 55, 57), had significant absorbance changes within a 5–40 h period, while only a few (No. 1, 32, 37, 58) exhibited most of their spectrum transitions only after > 40 h of incubation. The rest of the flavones had minor spectra changes during incubation that are reflected in slight transitions of the maxima positions (No. 30, 45, 54, 56, 60, 62) or a decrease in the magnitude of the maxima in the 380–420 nm region (No. 21, 43).

Examining the effect of non-oxidized flavones reveals that only the presence of luteolin (Figure 2A,B No. 56) slightly increased the aggregation halftime of insulin (Appendix A) while not affecting the fluorescence intensity. Other flavone relative halftime and ThT fluorescence intensity did not change, except for a few cases, where they even decreased the aggregation halftime (Figure 2A No. 10, 21–22, 48, 53, 54, 59, 61–63). However, once flavones were oxidized, many of them displayed substantial inhibitory potential. Some flavones (Figure 2A,B No. 31, 59, 63) increased the aggregation halftime more than five-fold, which correlates with the ten-fold elevated fluorescence intensity (compared to the control sample). In most cases, oxidized flavones inhibited insulin aggregation, except for a few (Figure 2A,B No. 1, 30, 32, 37, 46, 54–55, 58) that did not possess such properties, as neither ThT fluorescence intensity nor halftime changed compared to the previously tested non-oxidized forms. A completely different effect was seen on Aβ_42_ aggregation. Here, the fluorescence intensity (Figure 2D) was diminished in all cases, except for four flavones (Figure 2D No. 1, 30, 32, 37) which seem to have had no impact on either protein aggregation process, while several oxidized compounds (Figure 2D No. 22, 31, 52, 59) showed reduced intensity values ranging from 93% to 98%, which also reduced the aggregation rate. Despite the fact that most oxidized flavones inhibited insulin aggregation, only thirteen (Figure 2C No. 22, 31, 38, 43, 46, 48, 51, 52, 56–57, 59–60, 63) appeared to increase Aβ_42_ relative halftime and only three (Figure 2C No. 22, 31, 52) slowed the aggregation by at least 50%.

The flavone autoxidation experiment described above allowed us to evaluate the effect of oxidized flavones on protein aggregation. Nevertheless, not all compounds may undergo structural changes in the reaction mixture; thus, an additional number of flavones were incubated at the experimental conditions to evaluate whether UV-vis spectrum changes occur. Every tested flavone maintained the absorbance of Band I and Band II, with no major changes in the tested region (Figure 3). However, spectra of many compounds exhibited intensity changes with no shape or maximum transitions (No. 3, 6, 8, 9, 12, 13, 14, 19, 23, 26, 33, 47) that may be related to the solubility of each molecule, especially when the change occurred between the first two scans.

An identical experiment was conducted with the second set of flavones to evaluate their influence on insulin (Figure 4A,B) and Aβ_42_ (Figure 4C,D) aggregation processes. Here, similar results were observed, where most of the non-incubated and incubated flavones did not inhibit insulin aggregation, yet some increased its rate (Figure 4A No. 2, 3, 7, 8, 9, 12, 13, 15, 16, 18, 19, 20, 49, 50). The majority of flavones did not affect Aβ_42_ aggregation as well. However, a significant decrease in ThT fluorescence intensity was mostly evident for flavones with a higher number (Figure 4D No. 34–35, 41, 49–50), which represents THF and TeHF. In addition, dihydroxyflavones did not reduce the intensity value, except for no. 15. Three flavones (Figure 4C,D No. 5, 14, 16) that stand out appear to have altered the aggregation process by increasing the ThT fluorescence intensity and decreasing Aβ_42_ aggregation halftime.

Atomic force microscopy imaging was employed to observe whether fibrils were formed at the end of the Aβ_42_ aggregation experiment (when plateau was reached). Five samples were tested that represented the control sample (Figure 5A,B) and Aβ_42_ with incubated 2′,3′-DHF (Figure 5C,D), 6,2′,3′-THF (Figure 5E,F), 3,6,2′,3′-TeHF (Figure 5G,H), 3,6,3′,4′-TeHF (Figure 5I,J), 5,7,3′,4′,5′-PHF (Figure 5K,L). These particular compounds were selected due to their high impact on Aβ_42_ aggregation rate and bound-ThT. All samples with flavones revealed Aβ_42_ fibrillar aggregates on the mica, despite the fact that the surface was mostly covered by round-shaped oligomeric, very short fibrillar structures. Samples with 2′,3′-DHF, 6,2′,3′-THF, 3,6,2′,3′-TeHF, and 5,7,3′,4′,5′-PHF (Figure 5C,F,G,K) appeared to have clumps of fibrils with round-shaped oligomeric structures attached to them, leaving the area empty around this structure. This suggests that inhibition requires the binding of an active molecule to the protein or its oligomeric/fibrillar species. In order to further analyze AFM images, we measured the height of a hundred oligomeric structures or fibrils and compared their height distribution (Figure 5M). Structures formed with inhibitors had a dispersed height distribution, revealing that oligomeric structures may resemble clumped protofibrils. To understand this aspect more, the FTIR spectra of control Aβ_42_ fibrils and the sample with 2′,3′-DHF (when both samples reached a plateau in the ThT intensity) were recorded (Figure 5N). Samples for this experiment were prepared by using 10 kDa concentrator tubes that aided in changing the reaction solution to D_2_O. This method also eliminated monomeric species of amyloid-β. Notably, the FTIR spectrum of control fibrils exhibited the only major maximum at 1630 cm^−1^, typical for β-sheet structures, commonly found in amyloid fibrils, while the spectrum of Aβ_42_ + 2′,3′-DHF sample, had a less expressed β-sheet-related band at 1629 cm^−1^, and another broad peak at 1675 cm^−1^, which can mean the presence of substantial amounts of turns or different types of β-sheets. Unfortunately, the FTIR spectra could not be analyzed deeper; due to very low signal intensity, the signal-to-noise ratio was too high. It is necessary to note that, before spectra were normalized, the area of the amide I band of the sample with inhibitor was almost twice as small as the area of the amide I band of the control sample, leading to an assumption that less oligomeric and fibrillar species were present.

## 4. Discussion

The characteristics of insulin aggregation kinetic data show that 63 out of 64 tested non-oxidized flavones possess no anti-amyloid properties under the tested conditions (Appendix A), while most flavones that undergo the autoxidation process slow down insulin fibril formation. This is expressed in altered relative aggregation halftime. However, compounds also change the ThT fluorescence intensity (Figure 2A,B), which can be explained based on our previous report, where we show that insulin is capable of forming distinct fibril conformations in 20% acetic acid solution, with one exhibiting ~10-fold higher bound-ThT intensity values [41]. Increased fluorescence intensity is also observed using oxidized gallic acid [38], leading to a hypothesis that oxidized flavones redirect insulin amyloid formation.

Contrary results are seen during the Aβ_42_ aggregation process. Here, oxidized flavones led to a reduced ThT fluorescence intensity (Figure 2D and Figure 4D), and only 14 oxidized flavones (Figure 2C and Figure 4C) affected the aggregation rate. These diverse results introduce several potential explanations which may act simultaneously during the kinetic experiment. First, molecules that act as inhibiting agents should bind to monomers, intermediate oligomeric species, or aggregation nuclei to prevent the aggregation process [42]. Matos et al. revealed that quercetin, luteolin, and (+)-dihydroxyquercetin non-covalently bind to Aβ_42_ lysine residues [27] and Sato et al. displayed the mechanism where catechol-type flavonoids, namely (+)-taxifolin, autoxidize forming an o-quinone on the B-ring that covalently binds to the amino group of lysine [28]. Second, the fluorescence quenching is unavoidable when using ThT as the excitation and emission wavelengths overlap with the majority of oxidized flavones absorbance region (Figure 1 and Figure 3) and appear to form oxidized flavone-ThT interactions (as seen from differences in absorbance spectra, when the compounds are separate or together, Appendix A) that may lead to less bound-ThT on the fibril surface, reducing the fluorescence intensity even further (Appendix A). Therefore, most of the oxidized flavones (especially with more OH groups) suppress the fluorescence intensity in Aβ_42_ aggregation experiments. This effect has been observed when two dye molecules interact alone or in the presence of fibrils [30].

Taking a deeper look into the AFM images, we see a tendency for the formation of major clumps when Aβ_42_ aggregates with oxidized flavones, especially with 2′,3′-DHF (Figure 5C) and 3,6,2′,3′-TeHF (Figure 5G). This indicates that flavone derivatives bind to the surface of higher-level oligomeric particles as well as fibrils. While most of the mica is covered by oligomeric species, the AFM images may be analyzed, and it can be concluded that inhibitors redirect the aggregation pathway towards the arrangement of different structures. However, this explanation is just the tip of the iceberg, and a more revealing image is seen after a larger-scale analysis. The fibrillar clumps, which appear to be a combination of oligomeric structures and fibrils, consist mostly of aggregates present on the mica that is hardly found. Despite this, some oligomeric species that are found around these clusters led to the assumption that aggregation was partially stopped.

The main objective of this work was to understand the variety of flavones that may act as inhibiting molecules. There is a distinct correlation between the positions of hydroxyl groups, flavone oxidation, and inhibition of the insulin and the Aβ_42_ aggregation process. Adjacent OH groups have a tendency to increase the solubility compared to other flavones and enable the autoxidation process, which was seen via UV-vis absorbance spectral data (Figure 1 and Figure 3). Taking into consideration dihydroxyflavones, only four (5,6-DHF, 7,8-DHF, 2′,3′-DHF and 3′,4′-DHF) had an influence on the protein aggregation process. Surprisingly, 6,7-DHF does not autoxidize or affect protein aggregation. Despite this, the majority of hydroxyflavones that have neighboring hydroxy groups undergo oxidation leading to an enhanced inhibitory potential. This structural aspect is similar for 6,7,3′-THF and 5,6,7-THF, while 5,6,7,4′-TeHF and 6,7,3′,4′-TeHF tend to oxidize, potentially due to the additional hydroxy groups on the flavone B ring. 2′,3′-DHF appears to have the highest inhibition potential out of all tested flavones, which is then followed by 6,2′,3′-THF. This may resemble a close connection between structures and the autoxidation end products. Surprisingly, some of the flavonol derivates that do not have neighboring hydroxyl groups (3-hydroxyflavone, 3,5,7-THF, 3,7,3′-THF, 3,7,4′-THF, 3,5,7,4′-TeHF and 3,5,7,2′,4′-PHF) undergo autoxidation; however, these autoxidized molecules do not increase the Aβ_42_ aggregation time. This finding suggests a distinct autoxidation mechanism as well as different cinnamoyl system characteristics that are decisive for the developed anti-amyloid properties. Further, flavones with a higher number of hydroxyl groups that contain the aforementioned neighboring OH groups do autoxidize and inhibit insulin aggregation, but only some extend the Aβ_42_ aggregation time. These flavones can be categorized into two groups: 7,8-DHF derivatives (7,8,2′-THF, 7,8,3′-THF, and 7,8,3′,4′-TeHF) and flavones that have at least two hydroxyl groups on ring B (3,6,3′,4′-TeHF, 5,7,3′,4′-TeHF, 5,7,3′,4′,5′-PHF, 3,6,2′,4′,5′-PHF and 3,5,7,3′,4′,5′-PHF). Even though the number of effective inhibitors directly correlates with the number of OH groups on the molecule, the penta- and hexahydroxyflavone groups are far more complex. One probable scenario is that the flavone inhibitory effect is enabled by the appearance of particular molecular structures that form during the autoxidation process. These molecules should be structurally related, as the positions of OH groups on the molecule repeat, potentially leading to similar autoxidation mechanisms and products. While this study shows that the autoxidation of flavones leads to the formation of different structures, it is essential to note that due to this process, flavones may lose their initial characteristics, such as being inhibitors of AChE or anti-oxidants.

## 5. Conclusions

Taking everything into account, non-oxidized flavones do not inhibit the aggregation process of insulin or amyloid-beta, while their oxidized forms show potential against fibril formation. We also show that flavone autoxidation and inhibition are strictly related to the structure of the molecule and depend highly on the position of hydroxyl groups.

## Figures and Tables

**Figure 1 antioxidants-10-01428-f001:**
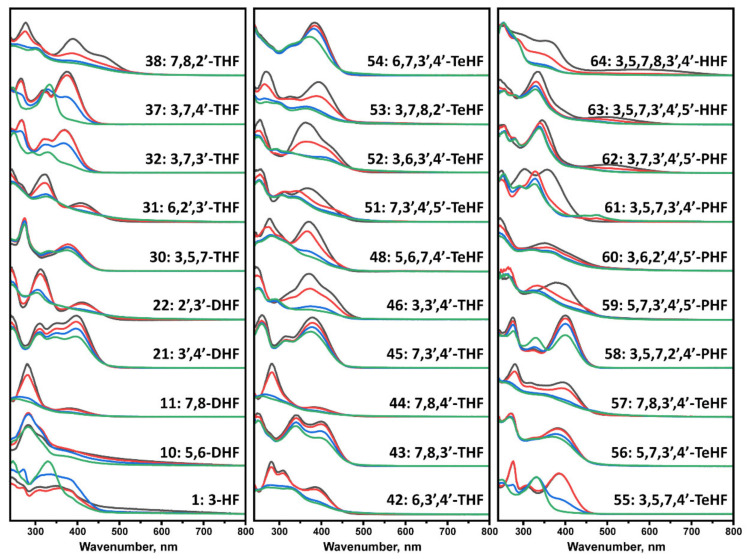
UV-visible absorbance spectra of flavones, recorded at 0 h (black), 5 h (red), 40 h (blue), and 100 h (green). Spectra were baseline corrected at 800 nm. Most of the flavone spectra experienced a significant change in the 250–450 nm region. In contrast, 21, 30, 43, 45, 54, 56, 60, and 62 experienced only a slight transition of maxima or decrease in the magnitude of the initial absorbance spectrum.

**Figure 2 antioxidants-10-01428-f002:**
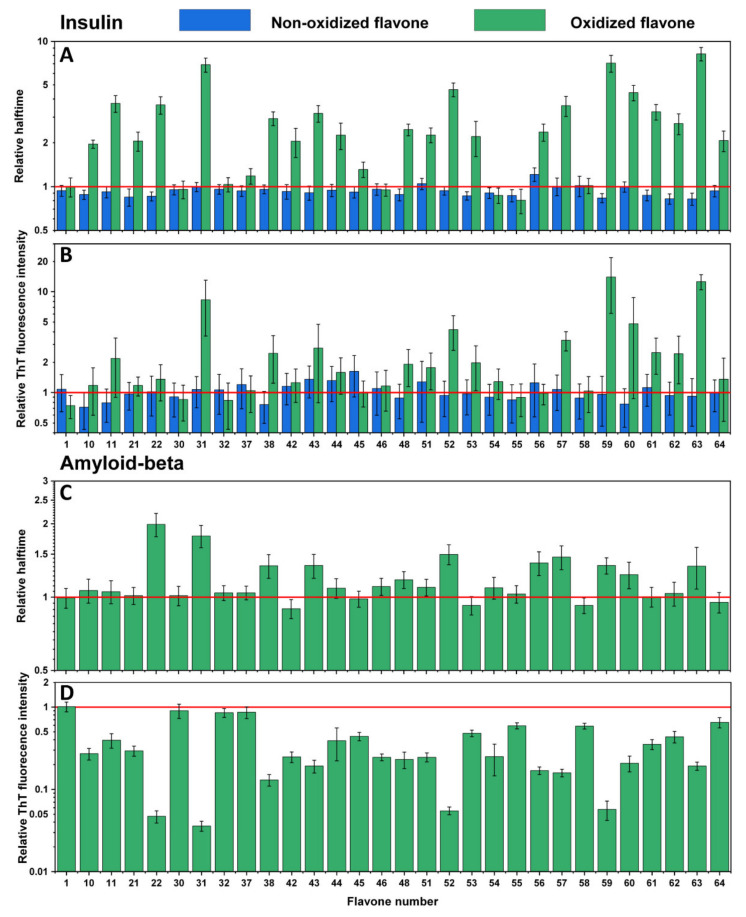
Effects of non-oxidized and oxidized flavones on insulin aggregation kinetics (**A**) and relative ThT fluorescence intensity (**B)**. Effect of oxidized flavones on Aβ_42_ aggregation kinetics (**C**) and relative ThT fluorescence intensity (**D**). Error bars are for one standard deviation (*n* = 4). None of the non-oxidized flavones, except 56, inhibited insulin aggregation; after the oxidation, more than half of the flavones showed an inhibitory effect, with 31, 59, and 63 having the most significant impact. Oxidized flavones 22, 31, 52, and 59 increased the relative halftime of Aβ_42_ the most, while 1, 30, 32, 37 did not affect the relative halftime nor the relative ThT fluorescence intensity.

**Figure 3 antioxidants-10-01428-f003:**
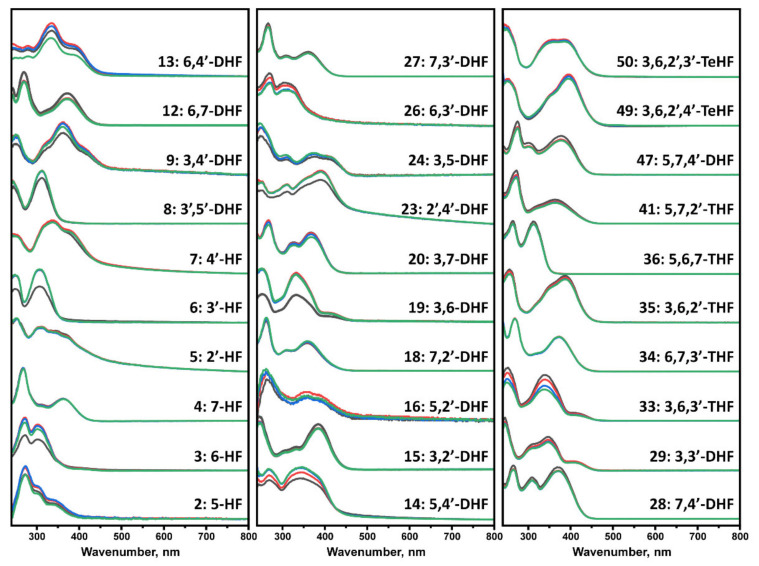
UV-visible absorbance spectra of flavones, recorded at 0 h (black), 5 h (red), 40 h (blue), and 100 h (green). Spectra were baseline corrected at 800 nm. Numbers 3, 6, 13, 14, 19, 23, 33 experienced the most significant decrease in the magnitude of the spectrum, while 4, 18, 27, 34, 36 had no notable change over the course of the experiment.

**Figure 4 antioxidants-10-01428-f004:**
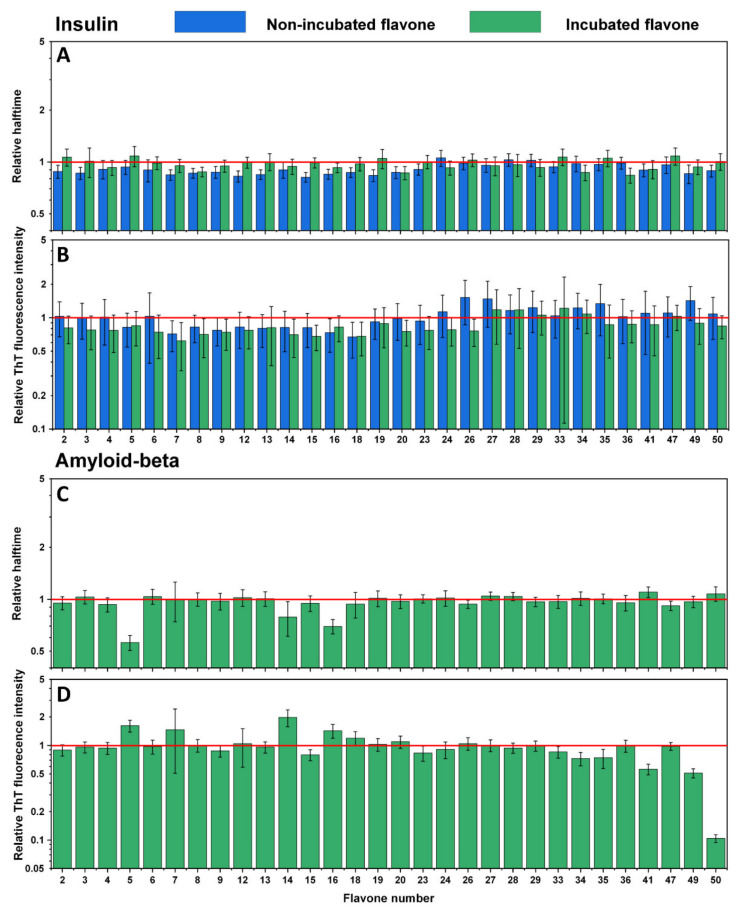
Effects of non-incubated and incubated flavones on insulin aggregation kinetics (**A**) and relative ThT fluorescence intensity (**B**). Effect of incubated flavones on Aβ_42_ aggregation kinetics (**C**) and relative ThT fluorescence intensity (**D**). Error bars are for one standard deviation (*n* = 4). The non-incubated and incubated flavones did not impact insulin and Aβ_42_ relative halftime, while incubated flavones 34, 35, 41, and 50 had the most significant impact on the relative ThT fluorescence intensity of Aβ_42_.

**Figure 5 antioxidants-10-01428-f005:**
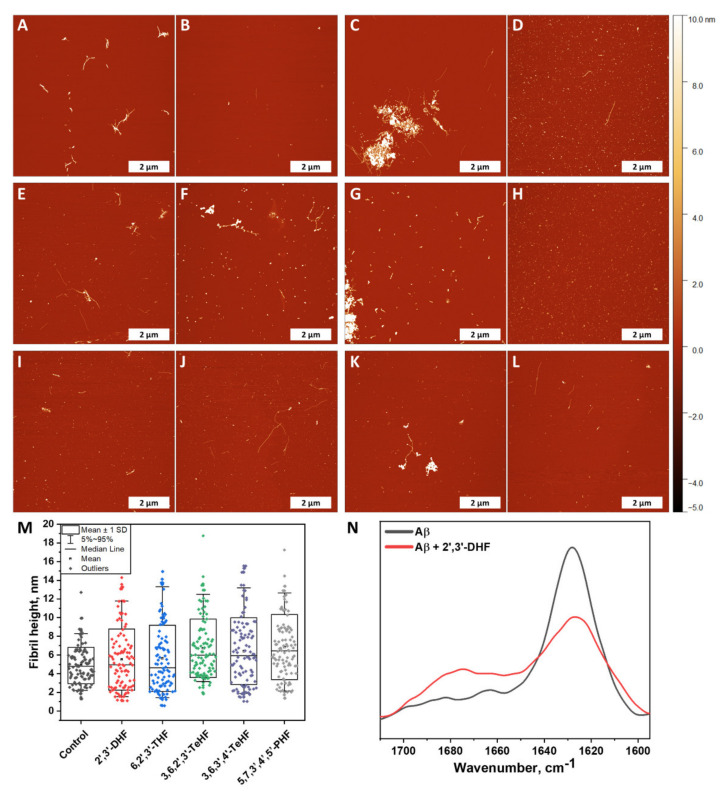
Atomic force microscopy images of Aβ_42_ formed without (**A**,**B**) and with 50 µM of oxidized 2′,3′-DHF (**C**,**D**), 6,2′,3′-THF (**E**,**F**), 3,6,2′,3′-TeHF (**G**,**H**), 3,6,3′,4′-TeHF (**I**,**J**) and 5,7,3′,4′,5′-PHF (**K**,**L**) flavones. Fibril and oligomeric species height distribution (**M**), where box plots indicate mean ± SD and error bars are in the 5%–95% range (*n* = 100). FTIR spectra (**N**) of Aβ_42_ fibrils formed alone and with 50 µM of 2′,3′-DHF. The AFM images of Aβ_42_ aggregates formed with all inhibitors showed a similar distribution in height and revealed round shape structures that were not present in the image of the control sample. The FTIR spectrum of the sample with 2′,3′-DHF had less expressed β-sheet-related band at 1629 cm^−1^ than the control sample.

## Data Availability

The data presented in this study are available in this manuscript.

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
