# Peer review of "Autoxidation Enhances Anti-Amyloid Potential of Flavone Derivatives"

_antioxidants, 2021, doi:10.3390/antiox10091428_

Round 1
Reviewer 1 Report
The authors investigated the inhibitory effect of 64 mono- and poly-hydroxylated flavones on the aggregation of amyloid-beta and insulin, which is commonly used for in vitro studies of amyloidogenic protein aggregation. The results suggested a great inhibitory effect of flavones on amyloid-beta and insulin aggregation.
This is an interesting study providing important insights into current knowledge on the pathophysiology and treatment of amyloidosis. It will contribute to the progress in exploiting natural anti-oxidative compounds and, therefore, attract broad range of readers. The manuscript is well written, and I do not have any critical comments.
Minor issues and suggestions to strengthen this manuscript are raised as follows:
- As this is an original article, the abstract should contain descriptions regarding background, methods, results, and conclusions. Methods and conclusions should be included. The results should be composed of concrete data even in the abstract.
- The authors mentioned the pathological hallmark of amyloidosis by focusing on oligomeric and fibrillar species without citation in the first paragraph of the introduction section.
- Findings obtained from figures should be provided in the legends for each figure.
Author Response
please find responses in the attached file

Reviewer 2 Report
The manuscript, titled to “Autoxidation Enhances anti-Amyloid Potential of Flavone Derivatives", discusses about the functionality of flavone derivatives against Aβ42 and insulin amyloidosis. This work reports the extensive analysis results showing anti-aggregation functionality of a large number of flavone derivatives whose chemical structures were systematically diversified, thus providing novel insights to understand the molecular mechanism of flavone derivatives. In particular, this work proved that autooxidation of flavone derivatives is critical to exert their anti-aggregation functionality. I think that this manuscript reports important implications to understand the general working mechanism of flavone-like molecules, and it may provide essential information to develop effective small molecule-based inhibitors of protein amyloidosis. I only have a few minor points to be considered for improving the manuscript:
- In the introduction, I think that it would be better if the authors explain why these flavone derivatives may work better than the antibody-based therapeutics, e.g., the recently-approved aducanumab.
- It would be better if the authors can include some explanations which chemical changes of flavone derivatives are expected to occur by auto-oxidation (particularly as a form of figure, such as in Fig. S1).
- The authors indicated that flavone derivatives may also work as AChEI. Could the authors include discussion about the possibility that the oxidation of flavone derivatives may affect their functionality as AChEI?
- I think that the current figures (Fig. 1~4) displaying the analysis results of various flavone derivatives (e.g., Fig. 1-4) is a bit complex and difficult to follow. Could this be re-arranged for better readability? For example, could the analysis results be categorized with the oxidation propensity and/or anti-aggregation functionality to insulin and/or Aβ42?
- The following experimental details in the method section need to be included:
(1) How oxidized (incubated) flavones were prepared, e.g., how long was the reaction mixture incubated before being used to test its anti-amyloid functionality?
(2) When was the ‘relative ThT fluorescence intensity’ measured? Were these measurements conducted consistently between the insulin and Aβ42 experiments?
Author Response

(The authors gave the same response as above.)
